# Epidemic Identification of Fungal Diseases in *Morchella* Cultivation across China

**DOI:** 10.3390/jof8101107

**Published:** 2022-10-20

**Authors:** Xiaofei Shi, Dong Liu, Xinhua He, Wei Liu, Fuqiang Yu

**Affiliations:** 1The Germplasm Bank of Wild Species, Yunnan Key Laboratory for Fungal Diversity and Green Development, Kunming Institute of Botany, Chinese Academy of Sciences, Kunming 650201, China; 2Guizhou Kangqunyuan Biotechnology Co., Ltd., Liupanshui 553600, China; 3School of Biological Sciences, University of Western Australia, Perth, WA 6009, Australia

**Keywords:** *Clonostachys solani*, *Diploöspora longispora*, ITS amplicon sequencing, fungal disease, morels cultivation

## Abstract

True morels (*Morchella*, Pezizales) are world-renowned edible mushrooms (ascocarps) that are widely demanded in international markets. *Morchella* has been successfully artificially cultivated since 2012 in China and is rapidly becoming a new edible mushroom industry occupying up to 16,466 hectares in the 2021–2022 season. However, nearly 25% of the total cultivation area has annually suffered from fungal diseases. While a variety of morel pathogenic fungi have been reported their epidemic characteristics are unknown, particularly in regional or national scales. In this paper, ITS amplicon sequencing and microscopic examination were concurrently performed on the morel ascocarp lesions from 32 sites in 18 provinces across China. Results showed that *Diploöspora longispora* (75.48%), *Clonostachys solani* (5.04%), *Mortierella gamsii* (0.83%), *Mortierella amoeboidea* (0.37%) and *Penicillium kongii* (0.15%) were the putative pathogenic fungi. The long, oval, septate conidia of *D. longispora* was observed on all ascocarps. Oval asexual spores and sporogenic structures, such as those of *Clonostachys*, were also detected in *C. solani* infected samples with high ITS read abundance. Seven isolates of *D. longispora* were isolated from seven selected ascocarps lesions. The microscopic characteristics of pure cultures of these isolates were consistent with the morphological characteristics of ascocarps lesions. *Diploöspora longispora* had the highest amplification abundance in 93.75% of the samples, while *C. solani* had the highest amplification abundance in six biological samples (6.25%) of the remaining two sampling sites. The results demonstrate that *D. longispora* is a major culprit of morel fungal diseases. Other low-abundance non-host fungi appear to be saprophytic fungi infecting after *D. longispora*. This study provides data supporting the morphological and molecular identification and prevention of fungal diseases of morel ascocarps.

## 1. Introduction

Wild morels are globally rare and precious edible fungi containing a variety of biological active ingredients [1,2,3,4,5]. The indoor cultivation of morels was not successful until the 1990s in the United States [6,7]. However, this indoor morel cultivation could not been operated at a large-scale due to its inconsistent harvesting yields. Fortunately, large-scale, outdoor morel cultivations up to 16,466 ha in the field have been commercialized in the 2021–2022 season across mainland China, based on the utilization of exogenous nutrients and successful domestication of easily cultivated varieties including *Morchella sextelata*, *M**. eximia* and *M*. *importuna* since 2012 [1,3]. Recent progresses have focused on species classification [8,9], genetic development [10,11,12,13], life cycle [2,14], nutritional metabolism [13,15,16] and multiomics [10,11,17,18,19,20,21,22,23,24,25,26,27], but less on cultivation management. Nevertheless, ~70% of the growers were unable to make a steady profit, mainly due to a lack of understanding in species cognition, mating type, growth and development, physiology and nutrition, and field management. In addition, the prevention and control of diseases is a key link to promote morel cultivation in the field.

The cultivation process of morels is mainly dependent on growth substrate, and an open environment makes their ascocarps susceptible to pests and to bacterial and fungal diseases in the field [28]. Pests, such as moth larvaes, mushroom flies, springtail, slugs, etc. [28,29], often feed and/or inhabit ascocarps, which then become deformed or damaged, and simultaneously induce bacterial and fungal diseases. With a red and withered stipe, the ascocarp becomes sticky and malodorous after being infected by bacterial diseases, but no specific pathogenic bacteria have been identified. The obvious symptoms of morel fungal disease, particularly common under high temperature and high humidity, are white villous lesions and wilting and malformation of the ascocarps [28,30,31,32,33,34]. Approximately 25% of the cultivation area is almost completed damaged after being infected by fungal diseases. In addition, some fungal diseases, such as *Fusarium* and *Alternaria*, also have potential risks to food safety [35,36]. Fungal diseases have become one of the most important concerns to a sustainable morel cultivation.

For *M. importuna,* cobweb disease was caused by *Cladobotryum protrusum* [30], pileus rot by *Diploöspora longispora* [31], stalk rot by *Fusarium incarnatum*—*F. equiseti* species complex [32], and both cap and stalk morbidity by *Paecilomyces penicillatus* [33], while in *M. sextelata* rot was caused by *Lecanicillium aphanocladii* [34]. In general, quicklime is widely used to sterilize and control pathogens during morel cultivation [28,29]. However, at present there is no epidemiological analysis on the fungal diseases of the morels.

To understand the status of morel fungal diseases, ascocarp samples with symptoms of fungal diseases were collected from major morel cultivation areas across China. To explore the prevention and control strategies of morel fungal diseases on a large scale, the major pathogens of fungal diseases in these morel samples were determined by combining a microscopic examination, ITS (internal transcribed spacers) amplicon sequencing technology and a pure culture identification.

## 2. Materials and Methods

### 2.1. Samples Collection and Tests

Sample collection (Table 1) was carried out in the main cultivation areas of morels in China (Figure 1). Ascocarps with serious disease were collected, and air-dried in room temperature or dried at 40–50 °C with drying equipment, and then deposited in a clean sample bag and preserved at 4 °C. According to the original disease status of each specimen, a 1 cm^2^ tissue block centered on the lesion was cut with a sterile scalpel for subsequent DNA extraction and ITS amplicon sequencing analysis.

### 2.2. DNA Extraction and Library Sequencing

A Power Soil^®^ DNA Isolation Kit (Catalog No. 12888, MoBio^®^, Carlsbad, CA, USA) was used for genomic DNA extraction according to manufacturer’s instructions. The ITS1 between 18S and 5.8S ribosomal rDNA region was selected for analysis using the primers ITS1F (CTTGGTCATTTAGAGGAAGTAA) and ITS2 (GCTGCGTTCTTCATCGATGC). First, a sequencing connector was added to the primer end, ITS1 PCR amplification was carried out, and then the PCR products were purified, quantified, and homogenized to form a sequencing library. The constructed library was subject to a library quality inspection. A PE250 double terminal sequencing was carried out with Illumina novaseq 6000 after a quality inspection. The original image data obtained by the high-throughput sequencing were converted into Sequenced Reads after Base Calling, and the results were stored a FASTQ (FQ) file, which contains the sequence information of reads and its corresponding sequencing quality information. The original sequencing data and sample information were submitted to NCBI public database with the bioproject ID as PRJNA864112.

### 2.3. Data Analysis

Quality control was carried out on the original sequences (Trimmomatic v0.33), and the high-quality sequence was obtained through low-quality and length filtering. A Usearch (v10) software was used to combine clean reads of each sample with overlap [37]. UCHIME (v4.2) software was used to identify and remove chimeric sequences and obtain the final valid data [38]. In phylogenetic or population genetics research, the clustering standard is artificially set for a taxon (isolates, species, genus, grouping, etc.) to facilitate analysis. Usearch software was used to cluster reads at a similarity level of 97.0% and obtain OTUs. High quality sequences were clustered, differentiated as operational taxonomic units (OTUs), and identified to species using UNITE OUT database. A QIIME 2 software was used to conduct taxonomic analysis on samples at each classification level based on the results of feature analysis, and the community structure map, species clustering heat map, phylogenetic tree and taxonomic tree of each sample at the taxonomic level of phylum, class, order, family, genus and species were obtained [39]. T-test was used to evaluate the difference of alpha diversity index of samples from different collection sites. Alpha diversity analysis was used to study the species diversity of individual samples. Ace, Chao1, Shannon and Simpson indexes of each sample were calculated, and sample dilution curves and graded abundance curves were plotted. Beta diversity analysis was used to compare the differences in species diversity (community composition and structure) among different samples. The UPGMA tree, NMDS analysis, sample clustering heat map, sample PCA, PCoA map (with grouping information) and boxplot based on multiple distances were obtained according to the distance matrix. Through functional predictive analysis, the gene function or phenotype of samples can be predicted, and the abundance of functional genes or phenotypes can be calculated.

### 2.4. Isolation and Identification of Pathogens

In a sterile environment, the pathogen in the lesion was carefully selected with a needle. It was then streaked on CYM (2 g·L^−1^ yeast extract, 2 g·L^−1^ peptone,1 g·L^−1^, K_2_HPO_4_, 0.5 g·L^−1^ MgSO_4_, 0.46 g·L^−1^ KH_2_PO_4_, 20 g·L^−1^ glucose, 20 g·L^−1^ agar) solid culture medium. After static incubation at 21 °C, the germinated single colony was purified. Morphological identification, genomic DNA extraction, ITS full sequence (ITS1 and ITS4 primers) amplification and sequencing, and phylogenetic analysis were performed to determine the taxonomic status of pure culture isolates. The above-mentioned isolated pathogenic filaments were carefully selected with a needle and preserved in cryovials filled with 15% glycerol solution. Morphological characteristics of mycelia, asexual spores and chlamydospores were observed with a biological microscope (Lecia M205, Germany). For analyzing ascocarps lesions, the pathogen in the lesions was carefully selected, and the samples were mounted in 15% glycerol solution on microscope slides. The morphological characteristics of mycelia, asexual spores, sporulation structure and chlamydospores were also observed under a biological microscope. The phylogenetic analysis was performed by ClasterX and Iqtree software.

## 3. Results

### 3.1. Epidemic Characteristics of Samples with Pathogen Contamination

Pathogen samples were collected from 32 sites from 18 provinces, covering almost all current morel cultivation areas in China. The maximum linear sampling interval was 3300 km from southeast to northwest China (D1094) and 2900 km from southwest to northeast of China (D1095) (Table 1, Figure 1). 52% of the fungal diseases were on the morel cap (Table 1). The lesions in ascocarps were conspicuous and white villous or slightly powdery (conidia), and the ascocarps became atrophied or deformed as the disease progressed (Figure 2). Approximately 39.4% of ascocarps showed a simultaneous onset of stipe and cap symptoms, and the lesion characteristics of stipes were as those of caps. Taken together these lesion characteristics, the morel disease is often categorized as a mycotic wilt.

A small number of ascocarp (9.1%) were mainly affected by the stipe, and the lesion was characterized by invading the whole ascocarp from bottom to top, eventually resulting in an obstruction of nutrition transportation and then ascocarp death (Figure 2). The fungal pathogen is mostly centered on the exogenous nutrition bag and its surroundings. The mycelium of the pathogen is as thick as a spider silk that often called spider web disease for morel [30] (Figure 2H,I). When soil temperature was greater than 18 °C and air humidity was greater than 80%, the propagation speed of fungal pathogens was faster, then the morel ascocarps were invaded (Figure 2). Because the conidia of mycotic wilt have the nature of diffusion and transmission, they can quickly spread to the entire planting filed, particularly under >25 °C and >85% air humidity.

### 3.2. Control the Quality of Sequencing

A total of 7,683,987 pairs of reads were sequenced, and a total of 7,635,200 clean reads were generated after double-end reads quality control and splicing, with at least 79,048 clean reads generated per sample and an average of 79,533 clean reads generated. The quality of Q30% base of clean reads was the lowest of 98.52%, with an average of 99.20%, indicating high sequencing quality to experimental requirements. The average length of clean reads was 244.9 bp, with the shortest 232 bp and the longest 263 bp, which was consistent with the matching interval of ITS1 sequencing (Appendix A).

### 3.3. Analysis of Alpha- and Beta-Biodiversity

A total of 44 OTUs were obtained from 32 samples (Figure 3A). Among them, D1037A had the least (14), and both D1017b and D1017c had the maximum (33) (Figure 3A). Appendix A showed differences in OTUs among different biological duplicate isolates. Results showed no significant differences between different sampling sites (Figure 3A). For example, ACE index ranged from 20.5 to 34.9, with an average of 26.73 ± 3.08, indicating that there was minor difference in biodiversity between sample collection sites. 99.99–100.00% of the alpha diversity coverage indicated that the sequencing results represented the presence of all microorganisms in tested samples (Appendix A). Similarly, the dilution curve indicated that the sequencing data reflected the species diversity in these samples (Figure 3B). Beta diversity showed that most samples had a certain degree of similarity, and a higher similarity between samples from the same collection site (Figure 3C).

### 3.4. Diversity of Pathogen Species of Ascocarps

In the annotation process of pathogen classification, *P. penicillatus* and *D. longispora* had extremely high syngenicity in ITS sequences (similarity > 99%, the specific classification status of these species remains to be analyzed) (Figure 4).

The morphology of the was obviously different: (*D. longispora* asexual spores were oval, 0–3 diaphragms, no asexual sporoderm structure, spore production in series, while *P. penicillatus* asexual spores were elliptic, without diaphragm, with distinct sporogenic structure and bottleneck [31,40]. Since UNITE database (Fungi ITS, Release 8.0, https://unite.ut.ee/, accessed on 20 January 2022) has not yet included the ITS sequence information of *D. longispora*, to avoid errors, the sequences of *P. penicillatus* in UNITE local database (SH194961.07FU_AY624194_refs, SH102264.07FU_EU553300_reps and SH013726.07FU_AY624194_refs) was corrected as *D. longispora* in this study. 

A total of 39 species were identified, which were assigned to Ascomycota (98.42%), Basidiomycota (0.32%) and Mortierellomycota (1.26%), and 15 orders. The highest relative abundance were Hypocreales (80.89%), Pezizales (16.97%), Mortierellales (1.26%), Eurotiales (0.28%) and Tremellales (0.24%).

The top10 species with the highest abundance were *D. longispora* (75.48%), *M. sextelata* (16.18%), *C. solani* (5.04%), *Mortierella gamsii* (0.83%), *M. importuna* (0.79%), *Mortierella amoeboidea* (0.37%), *Penicillium kongii* (0.15%), *Sarocladium kiliense* (0.14%), *Trichothecium roseum* (0.09%) and *Plectosphaerella niemeijerarum* (0.09%)*. D.*
*longispora* was detected in all samples, and its percentage of reads was the highest in most (92.75%) of the samples (Figure 5 and Figure 6).

In addition to host morel, *C. solani* had the second highest detection rate and reads coverage ratio and showed the highest reads ratio in D1017 (Average = 84.03% ± 9.48%), followed by D1036 (Average = 54.52% ± 26.07%) (Figure 6). It also has a high detection rate in D1021 and D1020, which were 13.77% and 8.65%, respectively. Only 2 of the 96 samples did not detected *C. solani*. The *Mortierella gamsii*, with the fourth highest abundance, only showed a high ratio of reads in some regions or specimens, such as 11.54% on average in D1026 specimens, 9.07%, 2.69% and 2.34% of D1003, D1017 and D1039 samples (Figure 6).

### 3.5. Microscopic Examination of Lesion and Pure Culture Analysis of Pathogeic Fungi

The analysis of ITS amplicon sequencing showed that *D. longispora* and *C. solani* were the two main pathogenic fungi of morels. Pathogens of all samples were examined under microscope, and lesions of 7 samples were isolated, pure cultured and further examined. The microscopic examination of the lesions showed that most of the lesions showed consistent microscopic morphological characteristics, the hypha of the pathogen was transparent, colorless, 2–4 μm diameter in septum, many conidia which were cylindrical, long, dumbbell to pod- shaped, 21.14 ± 3.08 (SD) μm × 4.89 ± 0.46 μm, and most of them contained one septum, and a few contain 2–3 septa. There was no conidiophore, and most samples did not detect the sporogenic structure of asexual spores (Figure 2J,K). In the pure cultures of the pathogenic fungi, obvious chains of long, septate segmented asexual spores were observed (Figure 2I). At the same time, elliptic conidia could occasionally be detected in the lesions (Figure 2K,L).

In this study, the pathogenic fungi were further isolated and identified from seven samples (Figure 4). The fungal morphology from the pure culture was white villous mycelia, irregular colony edges and irregular concentric rings (Figure 2G). Microscopic observation showed that there were tandem septoconidia, and the size and morphology of the spores were consistent with the main asexual spores on morel ascocarps lesion. Spherical or nearly spherical tandem chlamydospores can be easily observed in the pure culture, but it is difficult to be observed in lesions (Figure 2H). In the D1036, a large number of small elliptic, long elliptic or renal conidia were detected in addition to a small number of cylindrical sepidia of *D. longispora* (Figure 2L). The small elliptic conidia were 3.59 ± 0.28 μm × 6.58 ± 0.86 μm, without septa, transparent, colorless and bottleneck sporulation (Figure 2L). The long elliptic or renal conidia were slightly larger than the small elliptic conidia, 3.88 ± 0.27 μm × 9.11 ± 1.47 μm, without septa, transparent and colorless. These morphological characteristics are consistent with *Clonostachys* sp.

## 4. Discussion

### 4.1. Microorgansims Endemic to Cultivation Sites Are the Likely Source of Morchella Pathogens

Greenhouse cultivation of morel relies on exogenous nutrient bags that provide nutrition for sexual reproduction [1,15,26,41]. A large number of fungi or bacteria exist in the exogenous nutrient bags [15,26,41]. Although their abundances are not high, the *Paecilomyces* genus appears especially in the middle and late stages of cultivation [26,41]. The putative fungal pathogens of *Mortierella*, *Sarocladium*, *Penicillium*, *Plectosphaerella* and *Trichothecium* detected in this study were also proliferated in the middle and late stages of exogenous nutrient bags [41]. Tan et al. (2021) analyzed the soil microbial community of morel fruiting and non-fruiting cultivation sites in the large scale in the spring of 2020 in Chengdu Plain, southwest China, and found that the mushroom cultivation soils had more diverse fungal and bacterial communities, and the non-fruiting cultivation sites tended to have a high proportion of one dominating fungus such as *M**ortierella*. *Penicillium* was relatively common in both fruiting and non-fruiting fields (127 out of 128 soil samples were detected, and the average relative abundance was more than 2.90%) [42]. The abundance of *Penicillium* in non-fruiting field was higher than that in mushroom field [42]. Even when *M. extelata* was cultured in an artificially constructed culture medium, a 0.37% average abundance of *Penicillium* was detected, and was lowest at the beginning of sowing, peaked at 45 and 90 days after sowing, and decreased slightly at the harvest stage [15]. Yu et al. (2022) also pointed out that there were more *Penicillium* fungi hidden in soil which exhibited low or no mushroom yields [26]. The large-scale outbreak and invasion of the fungal disease of *D. longispora* on *Morchella* in 2–3 days under high temperature and humidity also showed that they were originally hidden in the cultivation lands. Therefore, it can be inferred that there is a transmission route of *D. longispora*, the original *D. longispor**a* exists in soil and air during the cultivation of *Morchella*, which can be rapidly proliferated under a nutrition enriched condition during the cultivation, then be gradually transported and invade the ascocarp and then spread to anywhere in the field presumably via infectious airborne conidia. Combined with the data of Tan et al. (2021) and Yu et al. (2022) [26,42], a large scale infection ascocarps or non-fruiting could occur when *D. longispor**a* (*Paecilomyces* genus) had rapidly reproduced in the morel cultivated field.

### 4.2. Morels Contain Complex Microbial Communities on Their Ascocarps

The microbiome of *M. rufobrunnea* at different stages of indoor cultivation revealed a total of 153 OTUs, and *Gilmaniella* was the dominant genus in soil where the ascocarp was successfully produced, while *Cephalotrichum* was the dominant genus in soil without mushroom growth [24]. Compared with the results of Longley et al. (2019), in the top 10 high abundance fungal species (except the host) detected in this paper, the relative abundance of *Mortierella* is higher (5.23%), while other fungi such as *Paecilomyces* (OTU_103), *Clonostachys* (OUT_137), *Penicillium* (OTU_44, OTU_75 and OTU_111), *Plectosphaerella* (OTU_58 and OTU_22) have been detected, but the relative content is very low [24]. However, *Sarocladium*, *Vishniacozyma*, *Trichothecium* and *Aspergillus* that were detected during the indoor cultivation of *M. rufobrunnea*, were not detected or their abundance in *M. sextelata* was very low in this present study [24].

Except for the host, the soil fungal community of the development stage of *M. sextelata* mushroom from Qujing, Yunnan, *China* was composed of 15.6% *Physiophora* sp. and 5.3–8.7% *Mortierella* [25], while these two fungi were not detected or their abundance was very low in the present study. It may be that the microbial communities of different *Morchella* hosts are different, or the main pathogens of the disease inhibit the growth of other saprophytic fungi [25].

### 4.3. Diploospora Longispora Is Likely the Main Culprit of Fungal Disease in Morchella Ascocarps

*Fusarium incarnatum-F. quiseti* complex caused the stipe decay of *M. importuna*, which only affected the stipe [32]. In this paper, *F. quiseti* was detected from 66 out of 96 samples but with low abundance, and the total abundance of all samples was only 0.03% (Figure 5). Two *Fusarium* sp. strains (*Fusarium* sp. 1 and *Fusarium* sp. 2) were from the diseased morel ascocarp, but pathological experiments showed that they did not cause the expected white mold disease that should not be the pathogenic of *M**. importuna* [33]. The author speculated that *Fusarium* might be the saprophytic pathogen that grew in the later stage of the ascocarps infected by *Penicillium* or *Paecilomyces* [33]. The diseased parts of the host samples detected in this paper cover the individual cap, stipe and the whole ascocarp, but no *Fusarium* sp. disease was detected, which showed that although *Fusarium* sp. could infect morels to some extent, causing the occurrence of rot disease, it should not be the main pathogen in large-scale cultivation and production.

In 3/20 morel plantations, there was a 2% to 5% incidence of cobweb disease among morel ascocarps in Tai’an, Shandong, east China [30]. At the initial stage of the disease, the arachnoid hyphae spread on soil surface and began to grow upward after contacting the stipe base until they infected the whole ascocarp, which became soft after infection. *Cladobotryum protrusum* was verified as one of the fungal pathogens [30]. The three pure stipe and 13 both cap and stipe infected samples from most *Morchella* cultivation sites across China were detected in this paper, however, no *Cladobotryum protrusum* had been detected, which might relate to a low incidence of *C. protrusumor*, its regional disease.

*Lecanicillium aphanocladii* is a newly reported fungus causing the rot of *M. sextelata* ascocarps [34]. However, the pathogen (*L. aphanocladii*) was not detected in this study. *L. primulinum* belonging to the same genus of *Lecanicillium* was detected but with very low relative content (accounting for approximately 0.02% of all samples) (Figure 4), suggesting that *L. aphanocladii* has scarcely presented as a fungal pathogen in the China.

The analysis of ITS amplicon sequencing showed that *D. longispora* and *C. solani* were the two main pathogenic fungi of morels across China, while the commonly-reported *Fusarium* spp. [32], *Cladobotryum* spp. [30] and *L. aphanocladii* [34] were not detected. Spherical or nearly spherical tandem chlamydospores can be easily observed in fungal lesion of morel ascocarps and the pure culture (Figure 1H), which was consistent with the morphological characteristics of *D. longispora* identified earlier [31]. *Diploospora longispora* was detected in all samples, and its percentage of reads was highest (75.48%) in most of the samples. Meanwhile, the elliptic conidia could occasionally be detected in the lesions (Figure 2K,L), which could be other saprophytic fungi different from *D. longispora*, these morphological characteristics are consistent with *Clonostachys* spp. [43] and should correspond to *Clonostachys*
*solani*, the second most important pathogen detected by ITS amplicon. Unfortunately, the specimen could not be isolated in pure culture due to drying ascocarps, and should be confirmed by further pathological experiments. A variety of pathogenic fungi coexisted in each sample by ITS amplicon, while most of them were dominated by one (*D. longispora* in most samples or two (reads of *C. solani* accounted for 48.79% while *D. longispora* for 11.26% in D1036), which indicated that the pathogen of morel should be dominated by one or two pathogens, and the other pathogens should be fungi inhabiting in the ascocarps. Considering the influence of samples variations from different years (3 years) and different producing areas, it can be speculated that *D. longispora* should be the main pathogen of morel ascocarp fungal disease.

## 5. Conclusions

The amplicon analysis of 32 fungal diseased ascocarps from 18 provinces across China showed a high abundance of *D. longispora* in almost all the lesions, although other non-host fungi with low abundance were also detected in this study. Combined examinations of both the abiotic and biotic factors (morphology, in situ separation and gene sequencing) of Chinese morel cultivation, our results demonstrate that *D. longispora* has presenting as a major culprit of morel fungal diseases. Other low-abundance, non-host fungi, including *C. solani*, could threaten the natural growth of ascocarps and morel cultivation. This present study provides both theoretical and practical strategies (such as lime application) for the prevention and control of fungal diseases in Chinese morel production.

## Figures and Tables

**Figure 1 jof-08-01107-f001:**
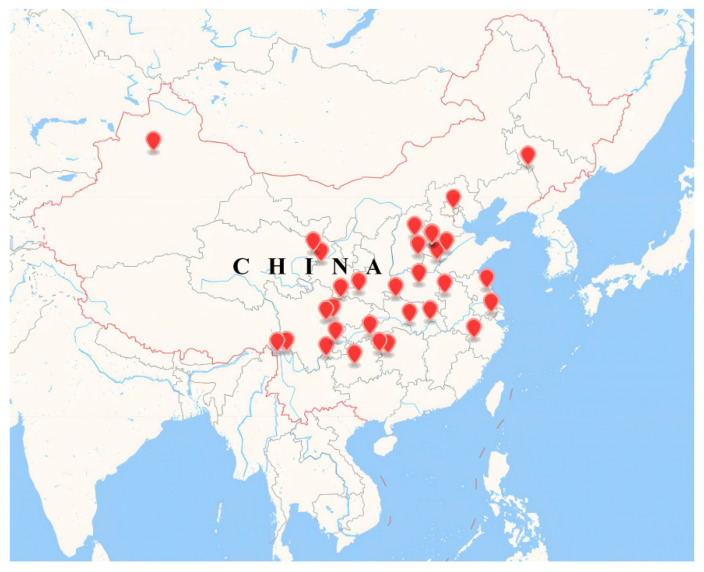
Sampling distribution of fungal disease samples from main morel cultivation farms across China.

**Figure 2 jof-08-01107-f002:**
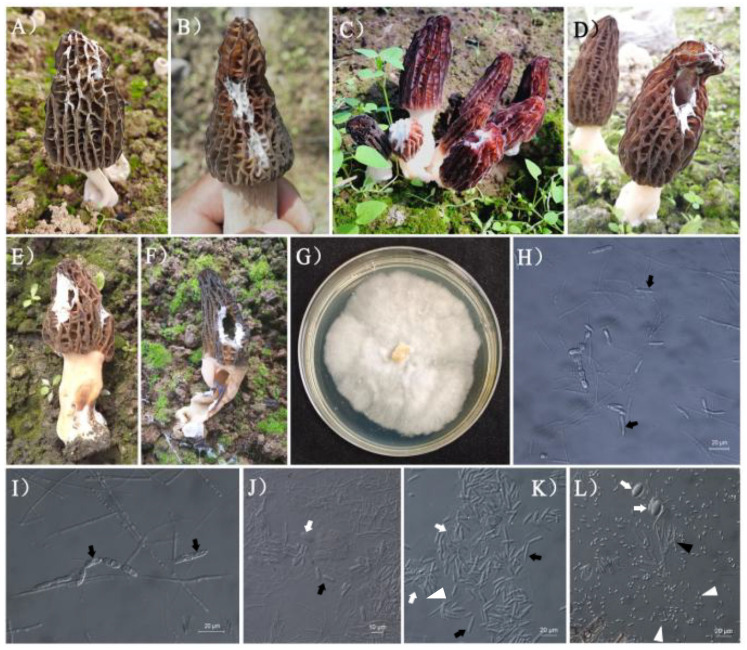
Morphological characteristics of fungal diseases of *Morchella*, Occurrences of fungal disease in morel ascocarps under outdoor cultivation. The diseased wilts and shrinks, and white hairy disease spots mostly occur on the cap, including the stipe (**A**–**E**); the fungal diseases lesion after 10 days of pathogen inoculation [31] on the healthy ascocarp (**E**,**F**); morphological characteristics of pure cultured pathogen cultured in a CYM medium at 24 °C for 14 days, with irregular, white, villous colonies and irregular concentric rings (**G**); morphological characteristics of chlamydospores and asexual spores of the D1092 isolate, chlamydospores are spherical and locate in the posterior wall, and asexual spores are oval shaped with 0–3 septum (**H**,**I**); morphological characteristics of pathogens in the focus of cap in D1012, D1095 and D1036 (**J**–**L**); the hollow arrow refers to the large elliptic ascospore of the host morel, the solid arrow refers to the long rod-shaped septate conida of *Diploöspora longispora* (**H**–**K**), the hollow triangle refers to the elliptic conida of *Clonostachys* (L), and the solid triangle refers to the conidiophore of *Clonostachys* (L).

**Figure 3 jof-08-01107-f003:**
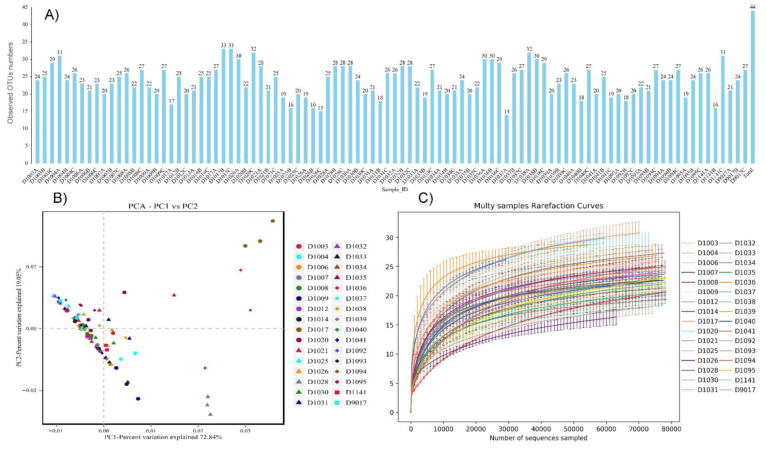
The number of the fungal OTUs (**A**) presented in the samples infected by morel diseases, (**B**) differences in community compositions as indicated by principial component analysis, and (**C**) rarefaction curves which can be used to compare species richness in samples with different amounts of sequencing data.

**Figure 4 jof-08-01107-f004:**
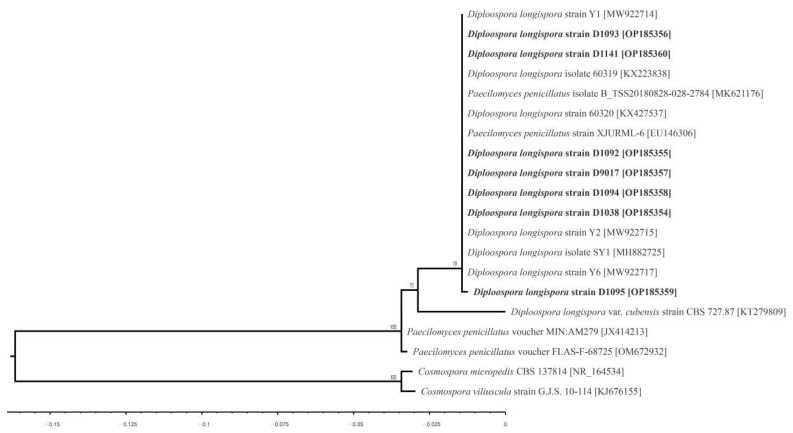
Molecular phylogenetic analysis by maximum likelihood method using ITS sequence of seven fungal lesion isolates. The bootstrap consensus tree inferred from 1000 replicates was taken to represent the evolutionary history of the taxa analyzed. New sequences generated from this study in bold, and the GenBank accession number was followed by species name.

**Figure 5 jof-08-01107-f005:**
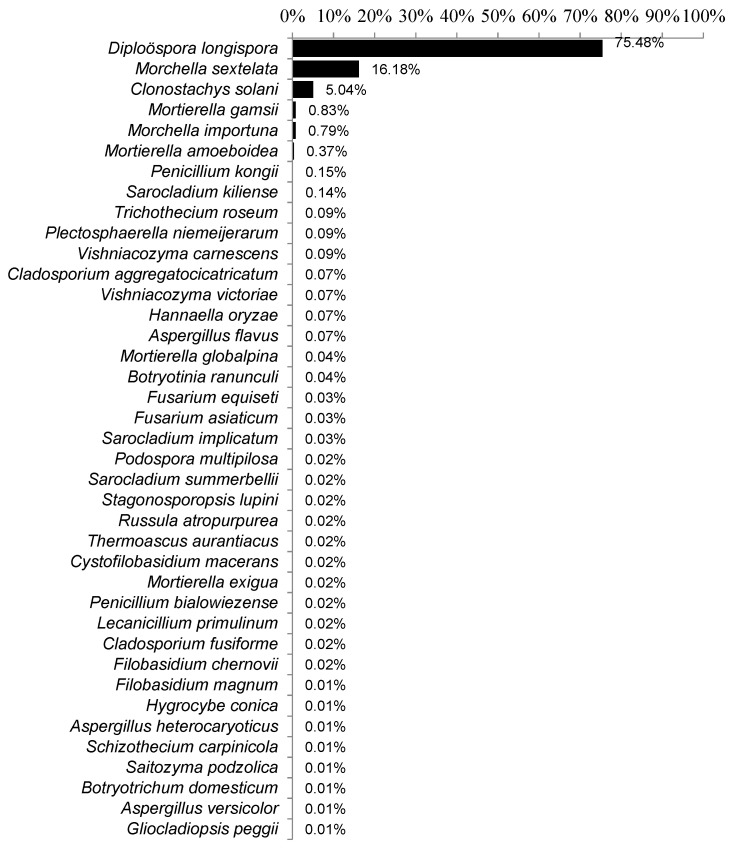
Variations in abundance or occurrence of putative pathogens in *Morchella* lesions across the morel cultivation across China.

**Figure 6 jof-08-01107-f006:**
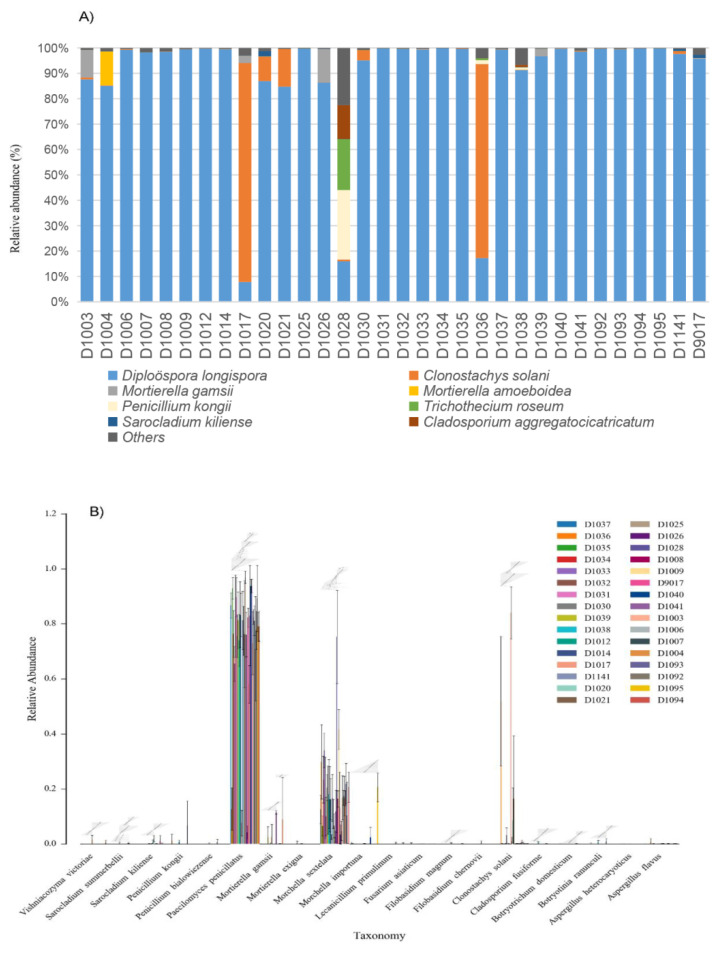
Microbial community composition (**A**) and abundance distribution (**B**) of fungi pathogens in 32 fungal lesions of *Morchella*.

**Table 1 jof-08-01107-t001:** Sampling location and *Morchella* specimen collecting across China.

Sample	Sampling	Sampling	Specimen	Sample	Lesion	Sporulation	Oval Asexual
Number	Location	Date	Collector	Status	Location	Structure	Spore
D1141	Yibing, Sichuan	20190108	Yun Hu	Air Drying	Stipe	No	No
D9017	Wuhan, Hubei	20190403	Wei Liu	Air Drying	Stipe	No	No
D1003	Jingzhou, Hubei	20200320	Zemin Qin	Bakeout	pileus	No	No
D1004	Pengshui, Chongqing	20200405	Zemin Qin	Bakeout	Pileus	No	No
D1006	Guiyang, Guizhou	20200317	Tao Liu	Bakeout	Pileus	No	No
D1007	Gongshan, Yunnan	20200328	Tao Liu	Bakeout	Pileus	No	No
D1008	Diqing, Yunnan	20200325	Tao Liu	Bakeout	Pileus	No	No
D1009	Boyang, Anhui	20200331	Zemin Qin	Bakeout	Pileus	No	No
D1012	Huaihua, Hunan	20210330	Haiqing Lin	Bakeout	Pileus	No	No
D1013	Huaihua, Hunan	20210402	Haiqing Lin	Bakeout	Pileus and Stipe	No	No
D1014	Zhaotong, Yunnan	20210320	Wenzhong Cai	Bakeout	Pileus	No	No
D1017	Jintang, Sichuan	20210301	Chaoqi Liang	Bakeout	Pileus	Yes	Yes
D1020	Qingchuan, Sichuan	20210325	Chaoqi Liang	Bakeout	Pileus and Stipe	No	Yes
D1021	Chongzhou, Sichuan	20210309	Chaoqi Liang	Bakeout	Pileus	No	Yes
D1025	Xingtai, Hebei	20210412	Wencheng Song	Bakeout	Pileus and Stipe	No	No
D1026	Tongren, Guizhou	20210318	Wencheng Song	Bakeout	Pileus and Stipe	No	No
D1028	Yancheng, Jiangsu	20210405	Zhenglei Xiang	Bakeout	Pileus	No	No
D1030	Xingtai, Hebei	20210408	Wencheng Song	Bakeout	Pileus	No	No
D1031	Shiyan, Hubei	20210406	Tianbing Zhao	Bakeout	Pileus and Stipe	No	No
D1032	Pingdingshan, Hunan	20210330	Yongfeng Man	Bakeout	Pileus and Stipe	No	No
D1033	Hanzhong, Shaanxi	20210403	Deming Huang	Bakeout	Pileus and Stipe	No	No
D1034	Puyang, Henan	20210412	Shoutao Gong	Bakeout	Pileus and Stipe	No	No
D1035	Liaocheng, Shandong	20210412	Shoutao Gong	Bakeout	Pileus	No	No
D1036	Suzhou, Jiangsu	20210405	Chi Song	Bakeout	Pileus and Stipe	Yes	Yes
D1037	Quzhou, Zhejiang	20210406	Wenhao Cui	Bakeout	Pileus and Stipe	No	No
D1038	Haidong, Qinghai	20210618	Wei Liu	Air Drying	Pileus and Stipe	No	No
D1039	Changzhi, Shanxi	20210510	Hong Zhang	Bakeout	Pileus	No	No
D1040	Jinzhong, Shanxi	20210520	Hong Zhang	Bakeout	Pileus and Stipe	No	No
D1041	Quzhou, Zhejiang	20210406	Wenhao Cui	Bakeout	Pileus and Stipe	No	No
D1092	Linxia, Gansu	20210427	Xiling Jia	Air Drying	Pileus	No	No
D1093	Beijing, Beijing	20210424	Xue Guo	Air Drying	Stipe	No	No
D1094	Shihezi, Xinjiang	20210403	Zhiyuan Cai	Air Drying	Pileus	No	No
D1095	Siping, Jilin	20210510	Guangxin Wei	Air Drying	Pileus	No	No

## Data Availability

The original sequencing data and sample information were submitted to NCBI public database with the bioproject ID as PRJNA864112.

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
