# Peer review of "Epidemic Identification of Fungal Diseases in Morchella Cultivation across China"

_jof, 2022, doi:10.3390/jof8101107_

Round 1
Reviewer 1 Report
This manuscript provides both a culture-dependent and metabarcoding approach to identifying possible causal agents causing various diseases on cultivated Morchella ascomata. I find this topic to be interesting and worthwhile considering the increasing cultivation of morels in China.
Overall, the manuscript requires effort to improve the clarity, grammar, flow and organization of ideas. I have included an attached PDF with comments and edits but I suggest using an English scientific proofreading service. I will include some points below:
- Section 2.2. DNA Extraction and Library Sequencing should be expanded upon with more details. For example, how was the PCR conducted (amplification conditions, reagents; describe or use reference)? How were samples pooled? What quality inspection was actual done? Provide references for the primers used in this study.
- I assume that OTUs were identified via GenBank nBLAST queries?
- 2.4. Isolation and Identification of Pathogen: I am not sure what is meant by a jumping needle. Do you mean the samples were mounted in 15% glycerol solution on microscope slides? I do not understand this.
- At times you refer to greenhouses: are morels also cultivated in greenhouses as well as outdoor fields?
- Very few reads did not pass quality control; was your QC strict enough?
- Fig. 2 H-L are a bit too dark to see properly, at least on my monitor. Consider adjusting the brightness/contrast/levels for clarity.
- Fig. 2: "after 10 days of fungus inoculation"; Were these experimentally inoculated or is this 10 days after the onset of symptoms in natural infections? If the former, please describe in methods and results or refer to the paper where experimental inoculations occurred.
- Fig. 2: I do not see a hollow arrow, solid arrow, or hollow triangle?
- en dash (–) between ranges.
- Fig. 4: Include more information, e.g.: scale bar? outgroup? species followed by isolate number with GenBank accession number in square brackets; new sequences generated from this study in bold; phylogenetic analysis used to generate tree (ML? Bayesian?)
- What is meant by "proletariat spore"?
- The morphological descriptions are sometimes confusing or unclear (see comments in PDF). For example "bottleneck sporulation" and "diaphragm" are unclear.
- I believe the reader will be confused by the identities of D. longispora and P. penicillatus. From my understanding, these are very closely related (conspecific?) species with virtually identical ITS sequences but different morphologies. Please clearly state at some point the taxonomy of these two species because at times they seem interchangeable; e.g., where is D. longispora in Fig. 5, is it also encompassed in P. penicillatus? If ITS cannot distinguish them than please state this and perhaps refer to it as a complex and/or change the figures so that the reader understands that the ITS sequences may be for P. penicillatus and/or D. longispora, it is just that ITS lacks resolution to distinguish these two taxa. Another example: you state that Paecilomyces was the most abundant genus (75.48%) which I assume represents D. longispora (75.48%). Is D. longispora actually a species of Paecilomyces or vice versa?
- Present dimensions as length x width not width x length; i.e.: 21.14 ± 3.08× 4.89 ± 0.46 μm. Also, is the ± standard error or standard deviation? If so please mention this.
- Rewrite 4.1 title for clarity. E.g.: Microorgansims endemic to cultivation sites are the likely source of Morchella pathogens
- Acremonium mortierella - do you mean Acremonium or Mortierella?
- Mushroom is used synonymously with ascocarp throughout the MS. Be consistent.
- Physiophora; do you mean Phialophora?
- 4.2: I don't see Phialophora in your Fig. 5 nor samples from Qujing in Table 1.
- Conclusions: What practical strategies are presented? Lime application? Do not overstate or be vague about the implications from this work.
- Throughout the discussion it can be somewhat unclear at first what results are from this study or from other studies.
- It would have been interesting to compare the mycobiomes on infected lesions versus asymptomatic tissues.
Overall, I think this manuscript is acceptable after minor revisions, however these revisions are not trivial and will require careful corrections and proofreading to ensure the study is easily understandable and well organized to the reader.

Author Response
Dear the Reviewer 1#
We are grateful for all your valuable comments. All the detailed suggestions have all been considered and revised.
We thanks again for your careful checkings in the pdf version, which greatly improved the overall quality of our manusript.

Reviewer 2 Report
The manuscript entitled ‘Epidemic Identification of Fungal Diseases in Morchella Culti-vation Across China’ by Shi et al. reports the ITS amplicon sequencing and microscopic examination of Morchella ascocarp lesions collected in 32 sites of 18 China provinces. Moreover, this study provides data support for the determination and prevention of fungal diseases of morel ascocarps. The manuscript is of interest and results are well discussed. I recommend the publication in Journal of Fungi after minor revisions as reported below
- Introduction:
The introduction is poor, general information on mushrooms is lacking. In fact, edible mushrooms are rich in enzymes with antifungal, antibacterial and cytotoxic activity. Please review the literature: e. g. https://doi.org/10.3390/toxins13040263 and https://doi.org/10.3390/toxins14020084. Moreover, further information on the nutritional properties of Wild morels should also be added. Please review the literature: e.g. https://doi.org/10.1080/10408398.2017.1285269.
- Materials and Methods:
Change ‘40–50°C’ by ‘40–50 °C’
please clarify this sentence: ‘The ITS1 region between 18S and 5.8S ribosomal rDNA’
change ‘g·l-1’ by ‘g·L-1’ check this in all manuscript.

Author Response
Dear Reviewer 2#,
We thank your detailed suggestions on the M&M. All the items have been revised. For the Introduction, we tend to keep the initial logic line since the suggested literatures were not stand in a line with our research background.
Thanks again for your valuable suggestions.
Reviewer 3 Report
Comments to jof-1934781
Title: Epidemic Identification of Fungal Diseases in Morchella Culti-vation Across China.
In this manuscript, the authors tried to explore the major pathogens of fungal diseases from major morel cultivation areas across China, by microscopic examination and ITS amplicon sequencing technology. This manuscript is well prepared. Only one suggestion is raised. The resolution and magnification of the photos showing morphology of putative pathogen (Figure 2, H-L) are too low r , so the morphology of cell spores are not clear. The photos are strongly suggested to be changed.
Author Response
Dear Reviewer 3,
We appreciate your recognition of our work. In the revised version, we have provided a new Fig. 2 with proper resolution and magnification of the photos.